# Impact of Sodium *N*-[8-(2-Hydroxybenzoyl)amino]-caprylate on Intestinal Permeability for Notoginsenoside R1 and Salvianolic Acids in Caco-2 Cells Transport and Rat Pharmacokinetics

**DOI:** 10.3390/molecules23112990

**Published:** 2018-11-16

**Authors:** Ying Li, Dandan Yang, Chunyan Zhu

**Affiliations:** Department of Pharmaceutics, Institute of Medicinal Plant Development, Chinese Academy of Medical Sciences and Peking Union Medical College, No. 151 Malianwa North Road, Haidian District, Beijing 100193, China; yli@implad.ac.cn (Y.L.); mayong@mohrss.gov.cn (D.Y.)

**Keywords:** notoginsenoside R1, salvianolic acids, Caco-2 cells, pharmacokinetics, sodium *N*-[8-(2-hydroxybenzoyl)amino]caprylate (SNAC), molecule polarity

## Abstract

For drugs with high hydrophilicity and poor membrane permeability, absorption enhancers can promote membrane permeability and improve oral bioavailability. Sodium *N*-[8-(2-hydroxybenzoyl)amino]caprylate (SNAC) is a new kind of absorption enhancer that has good safety. To investigate the absorption enhancement effect of SNAC on non-polar charged and polar charged drugs and establish the absorption enhancement mechanism of SNAC, SNAC was synthesized and characterized. Two representative hydrophilic drugs—notoginsenoside R1 (R1) and salvianolic acids (SAs)—were selected as model drugs. In vitro Caco-2 cells transport and in vivo rat pharmacokinetics studies were conducted to examine the permeation effect of SNAC on R1 and SAs. R1, rosmarinic acid (RA), salvianolic acid B (SA-B) and salvianolic acid B (SA-A) were determined to compare the permeation enhancement of different drugs. The MTT assay results showed that SNAC had no toxicity to Caco-2 cells. The transepithelial electrical resistance (TEER) of Caco-2 cell monolayer displayed that SNAC facilitated passive transport of polar charged SAs through the membrane of epithelial enterocytes. The pharmacokinetics results demonstrated that area under the curve (AUC) of RA, SA-B and SA-A with administration of SAs containing SNAC was 35.27, 8.72 and 9.23 times than administration of SAs. T_max_ of RA, SA-B and SA-A were also prolonged. The AUC of R1 with administration of R1 containing SNAC was 2.24-times than administration of R1. SNAC is more effective in promoting absorption of SAs than R1. The study demonstrated that SNAC significantly improved bioavailability of R1 and SAs. What’s more, the effect of SNAC on absorption enhancement of charged drugs was larger than that of non-charged drugs. The current findings not only confirm the usefulness of SNAC for the improved delivery of R1 and SAs but also demonstrate the importance of biopharmaceutics characterization in the dosage form development of drugs.

## 1. Introduction

There is considerable interest in delivery platforms that can improve the oral bioavailability of poorly absorbed drugs, because the oral route can improve patient compliance, and oral formulations can also reduce the costs associated with sterile manufacturing and use of healthcare professionals. The intestinal absorption of hydrophilic drugs is usually limited by their poor membrane permeability. Therefore, absorption enhancers (AEs) have often been studied to improve the absorption of these poorly permeable drugs. These AEs include surfactants, bile salts, chelating agents, fatty acids and coco-glucoside [1,2,3].

These different types of AEs have been shown to increase the intestinal absorption of poorly permeable drugs by various mechanisms. They do this by either opening epithelial tight junctions (TJs—the paracellular route), mildly perturbing the mucosal surface (transcellular permeation enhancement), or by non-covalent complexation with the payload. However, the AEs that are highly effective often cause damage to and irritate the intestinal mucosal membrane [4,5,6]. Therefore, effective and less toxic AEs must be developed and used in clinical practice. Sodium *N*-[8-(2-hydroxybenzoyl)amino]caprylate (SNAC) is a delivery agent that has been reported to enhance the permeability of a diverse spectrum of molecules, including proteins, such as insulin [7], calcitonin [8] and other macromolecules such as heparin [9]. SNAC has not been reported to be associated with significant disruption of the tight junctions, change in membrane fluidity, and toxicity, among others [10,11,12]. SNAC has both an absorption enhancement effect and low toxicity. It has been used in clinical studies [13,14].

Notoginsenoside R1 (R1) is an effective and structurally representative bioactive constituent of *Radix notoginseng*. Studies have reported that R1 has various activities such as protecting against cardiac hypertrophy in ApoE-/- mice [15], suppressing wear particle-induced osteolysis and RANKL mediated osteoclastogenesis in vivo and in vitro [16], inhibiting oxidized low-density lipoprotein induced inflammatory cytokines production in human endothelial EA.hy926 cells [17], attenuating amyloid-β-induced damage in neurons by inhibiting reactive oxygen species and modulating MAPK activation [18], regulating human colorectal cancer metastasis [19]. R1 is a saponin (Figure 1A), it has good water solubility, and the solubility and dissolution rate are not the main factors affecting drug absorption, low membrane permeability and high molecular weight are the main factors resulting in poor bioavailability [20,21], which restricts clinical use. Salvianolic acids (SAs) is effective components of *Salvia miltiorrhiza*, it have various activities such as antioxygenation [22], improving the cognitive function of rats with chronic stress-induced depression [23], cardioprotection [24], protecting brain endothelial cells after treatment with deprivation and reperfusion of oxygen-glucose [25], anti-emphysema [26], attenuating limb ischemia/reperfusion injury in skeletal muscle of rats [27], promoting functional recovery and neurogenesis via sonic hedgehog pathway after stroke in mice [28]. SAs are water-soluble components which are mainly composed of rosmarinic acid (RA, Figure 1B), salvianolic acid B (SA-B, Figure 1C) and salvianolic acid A (SA-A, Figure 1D). However, the bioavailability of salvianolic acid B in rats is only 2.3%. The extremely low oral bioavailability is mainly caused by poor biomembrane penetration [29], which has limited clinical application.

SNAC has been reported to act as an absorption enhancer (Figure 1E), but there have been few reports about the absorption enhancement effects on polar charged drugs and non-polar charged drugs. Polar charged SAs and non-polar charged R1 were thus chosen as model drugs. The effect of SNAC on SAs or R1 transport across Caco-2 cell monolayer in vitro was tested. The transepithelial electrical resistance (TEER) during the exposure to SNAC of Caco-2 cell monolayer was tested to study whether the absorption-enhancing effect of SNAC involved tight junction complex opening or not.

The pharmacokinetics of SAs or R1 in rats was tested and compared when SAs containing SNAC and SAs or R1 containing SNAC and R1 were administered. The work aimed to investigate and compare the impact of SNAC on the oral absorption of drugs with different polar charged properties. This will provide a research basis for drug selection using SNAC as absorption enhancer to improve drugs’ oral bioavailability.

## 2. Results and Discussion

### 2.1. Sodium N-[8-(2-hydroxybenzoyl)amino]caprylate (SNAC) Characterization

The ^1^H-NMR spectrum of SNAC is shown in Figure 2. The peak at *δ* (ppm) 9.876 (s, 1H) corresponded to the amide (-NH-). The phenolic hydroxyl (OH-) peak appears at 4.959 (s, 1H).

The peaks at 6.606, 6.624, 6.642 (t, 1H), 6.803,6.823 (d, 1H), 7.167, 7.185, 7.202 (t, 1H) and 7.804, 7.821 (d, 1H) ppm are the peaks of the hydrogens on the benzene, while the peaks at 1.287 (s, 1H), 1.487, 1.500 (d, 1H), 2.003, 2.021, 2.039 (t, 1H), 2.508 (s, 1H) and 3.266, 3.282, 3.298 (t, 1H) ppm are the peaks of the hydrogens on the alkyyl chain.

### 2.2. Cytotoxicity Study (MTT Assay)

The use of absorption enhancers is one of effective methods to improve poorly absorbed drugs’ oral bioavailability. However, the absorption enhancers that are highly effective often cause damage and irritate the intestinal mucosal membrane [30,31,32]. Therefore, to evaluate the toxicity of SNAC, MTT assay on Caco-2 cells was carried out. When the culture time was 24 h (Figure 3), the cytotoxicity of R1, SAs and SNAC appeared as concentration-dependent. When R1 was 400 μg·mL^−1^, the survival percentage was still above 90%. When SNAC was 200 μg·mL^−1^, the survival percentage was above 90%. When SAs was 50 μg·mL^−1^, the survival percentage was above 90%.

### 2.3. Caco-2 Cell Transport

Transport Papp of R1 or SAs containing SNAC across Caco-2 cell monolayer was shown in Figure 4A,B. The TEER showed no obvious change after transport across the Caco-2 cell monolayer, which suggested that the transport process did not involve the paracellular route. After incubation for 2 h and 24 h, TEER restored to its original state, which indicated drugs had no toxicity to cells. As was shown in Figure 4C,D, comparing with the Papp of R1 solution, the Papp of R1 containing SNAC had no significant difference. Comparing with the Papp of SAs solution, the Papp of RA and SA-B improved by 2.14-fold and 3.68-fold when cells were treated with SAs containing SNAC. The possible mechanism of AEs increasing oral bioavailability of poorly permeated therapeutic molecules includes the transcellular route, enhancing the transport across the epithelial membrane, the paracellular route or modifying the epithelial intercellular tight junctions [33,34,35,36]. SAs containing SNAC did not transport across Caco-2 cell monolayer by opening tight junctions, which indicated the pathway was transcellular route.

### 2.4. Chromatographic Condition

To obtain satisfactory chromatographic separations, UPLC conditions were optimized by selecting columns and adjusting the gradient elution for the separation of all of the compounds in this study [37]. A Phenomenex Kineter EVO C18 (2.1 × 50 mm, 2.6 μm) and an optimized gradient elution program was selected for the analysis of R1 and saikosaponin (IS) to achieve smooth baseline separation and produce the highest MS intensity and the best resolutions for most of the peaks tested (Figure 5A).

In order to represent the pharmacokinetic of the whole herb extract, it is better to select several effective ingredients as the target to investigate the pharmacokinetics. The content of RA, SA-B and SA-A in SAs is much higher than other components, therefore, RA, SA-B and SA-A were selected as the object molecules to study the pharmacokinetic of SAs.

To obtain satisfactory chromatographic separations, HPLC conditions were optimized by selecting columns and adjusting the gradient elution for the separation of all of the compounds in this study [38,39].

The chromatogram peaks of RA, SA-B and SA-A had good peak shapes, no interference from miscellaneous peaks, and a stable baseline, as shown in the following Figure 5B.

### 2.5. Validation of Analytical Methods

Calibration curves of R1 were linear over the concentration range of 1.97–964 ng·mL^−1^. Good linearity with a correlation coefficient r = 0.9997 was observed. Standard curve equation was y = 279.08x + 439.26, y represented peak area of R1/ peak area of IS, × represented concentration of R1/concentration of IS.

Calibration curves of RA, SA-B and SA-A were linear over the concentration range of 0.055–28.4 µg·mL^−1^, 0.19–12.16 µg·mL^−1^ and 0.058–29.9 µg·mL^−1^, respectively. Good linearity with a correlation coefficient r = 0.9996, r = 0.9998, r = 0.9996 was observed, respectively. Standard curve equation was y = 41536x − 3928.9 for RA, Y= 71971x + 13966 for SA-B, y = 59541x − 11430 for SA-A. y represented peak area of RA, SA-B or SA-A, x represented concentration of RA, SA-B or SA-A.

The absolute recovery of RA, SA-B, SA-A and R1 from plasma was determined to 80–120%. This method showed good precision, the precisions were measured to be less than 15%.

The limit of quantification (LOQ), defined as the lowest quantification concentration of R1, which could be detected in plasma was 1.97 ng·mL^−1^. The LOQ of RA could be detected in plasma was 55.47 ng·mL^−1^. The LOQ of SA-B could be detected in plasma was 190 ng·mL^−1^. The LOQ of SA-A could be detected in plasma was 58.40 ng·mL^−1^.

### 2.6. Pharmacokinetic Study

The concentration-time curve of R1 in plasma is shown in Figure 6A. The AUC chart of SA is shown in Figure 6B. The detailed pharmacokinetic parameters are shown in Table 1 and Table 2. It was evident that AUC of R1 with administration of R1 containing SNAC was 2.24-times administration of R1. However, AUC of RA, SA-B and SA-A with administration of SAs containing SNAC was 35.27, 8.72 and 9.23-times higher than administration of SAs. AUC of RA is larger than SA-B and SA-A. The possible reason was that the molecular weight of RA is smaller than that of SA-A and SA-B, SNAC could improve the absorption enhancement of RA more than SA-A and SA-B. T_max_ was prolonged with significant differences (*p* < 0.05), and the internal absorption time was prolonged. The results demonstrated that SNAC could improve the oral absorption of both R1 and SAs, but SNAC improved the oral absorption of SAs more than that of R1. The SAs showed significant slow release and enhanced absorption after adding the SNAC. C_max_ values of R1, RA, SA-B and SA-A were increased significantly with drugs containing SNAC as compared that of crude drugs-treatment.

There was a relatively lower enhancement effect on neutral R1, and a relatively higher promoting effect on ionic SAs. It may be because the ionic interaction makes it easier to form a complex that mimics the body’s natural biomolecular transport mechanisms [40]. The interaction helps to protect drugs from digestive enzymes and increases hydrophobicity so the moiety can passively permeate, after which the complex dissociates into the respective components. Thereby, SNAC can promote ionized molecules’ membrane permeability at a larger extent.

In this study, SNAC was added to improve the absorption of low permeability drugs. SNAC may work via transcellular pathway of cells, promoting transmembrane transport of SAs, significantly improving the absorption of SAs and enhancing bioavailability. The specific absorption enhancement mechanism needs further study.

## 3. Materials and Methods

### 3.1. Chemicals, Reagents and Animals

Saikosaponin A and notoginsenoside R1were bought from the National Institute of the Control of Pharmaceutical and Biological Products (Beijing, China). Salvianolic acid A was bought from Biopurify Phytochemicals Ltd. (Chengdu, China). Rosmarinic acid was bought from Shanghai Yuanye Biotechnology Co., Ltd. (Shanghai, China). Salvianolic acid B was bought from Shanghai Ronghe Pharmaceutical Technology Co. Ltd. (Shanghai, China). Salvianolic acid extract of *Salvia miltiorrhiza* was bought from Xian Xiaocao Botanical Development Co. Ltd. (Xian, China). *N*-[8-(2-Hydroxybenzoyl)amino]caprylate (SNAC) was provided by Shanghai Synmedia Chemical Co., Ltd. (Shanghai, China). Acetonitrile and methanol (UPLC-MS grade) were purchased from Fisher Scientific (Waltham, MA, USA). Heparin sodium was purchased from Beijing Yaobei Biological and Chemical Reagents Company (Beijing, China).

Sprague-Dawley rats, male, healthy, weighing 250 ± 20 g, were purchased from Vital River Laboratory Animal Technology Co. Ltd. (Beijing, China) for the animal experiments. Experimental animals were maintained in accordance with internationally guidelines for laboratory animal use, and the study was approved by the Ethical Committee of Experimental Animal Center of Institute of Medicinal Plant Development, Chinese Academy of Medical Sciences and Peking Union Medical College (No. SLXD-201807070364, Institute of Medicinal Plant Development, Chinese Academy of Medical Sciences and Peking Union Medical College, Beijing, China).

### 3.2. SNAC Synthsis

In brief, the raw material was dissolved in methanol, thionyl chloride was added dropwise into the solution that was then stirred overnight. The intermediate product was dissolved in dichloromethane and an accurate amount of trimethylamine was added. Aspirin was dissolved in dichloromethane and oxalyl chloride was added dropwise, followed by addition of triethylamine. Then dissolving in methanol, sodium hydroxide solution was added. Then the acidity was adjusted to excessive acid. Dissolving in the ethanol and then adding concentrated sodium hydroxide solution. Drying overnight and getting the final product.

### 3.3. ^1^H-NMR Characterization

SNAC was analyzed with DMSO-d6 as solvent using an Avance II (400 MHz) ^1^H-NMR instrument (Bruker, Dresden, Germany).

### 3.4. Cytotoxicity Study (MTT Assay)

Caco-2 cells were plated at a cell density of 4 × 10^5^ cells per well in 96-well plates and incubated at 37 ± 1 °C in an atmosphere of 5% CO_2_. After 24 h of culture, the medium was replaced with SNAC, R1 or SA. After 24 h, the medium was discarded and the wells were washed twice with hanks balanced salt solutions (HBSS). 200 μL MTT solution (0.5 mg·mL^−1^ in PBS) was added to each well, and incubated 4 h at 37 ± 1 °C for MTT formazan formation. Subsequently, the supernatant was carefully removed and the wells were washed twice with PBS. DMSO (200 μL) was added to each well and the plates were then mildly shaken for 15 min to ensure the dissolution of formazan crystals. The optical density values were measured by using MQX200 microplate reader (Bio-Tek, Shoreline, WA, USA) at wavelength 570 nm. Six replicates were read for each sample and mean value was used as the final result. The spectrophotometer was calibrated to zero absorbance using culture medium without cells.

### 3.5. Caco-2 Monolayer Transport

To investigate the influence of SNAC on the absorption properties of R1 and SA-A, SA-B and RA were used as an in vitro model of the gastrointestinal epithelium. To evaluate the transport, SNAC with R1 or SAs (1:1) were diluted with HBSS solution to a final concentration of 200 μg·mL^−1^ R1 or 50 μg·mL^−1^ SAs (SA-A: 6.07 μg·mL^−1^; SA-B: 8.31 μg·mL^−1^) as the test solutions. A 1.5 mL volume of sample was taken from the basolateral side at 2 h. Sample aliquots of 500 μL were mixed with 500 μL of methanol, shaken for 1 min using a vortex mixer, and centrifuged at 8000 rpm for 10 min. The supernatant was injected into the HPLC system for measuring R1, RA, SA-B and SA-A content. The apparent permeability coeffcient (Papp) was calculated according to the following equation: Papp = dQ/dt × 1/(AC_0_), where dQ/dt is the permeability rate, C_0_ is the initial concentration at the apical side, and A is the surface area of a monolayer. TEER was determined using a Millicell-ER system (Millipore Corporation, Bedford, MA, USA) before and after membrane absorption. After membrane absorption, the cells were washed three times with HBSS solution, complete media was added for 2 h or 24 h, and the TEER was then determined, the cell toxicity of test drugs and regeneration of cell membrane were also investigated.

### 3.6. Chromatographic System and Conditions

#### 3.6.1. Chromatographic System and Conditions of Notoginsenoside R1

The assay was performed on a Waters UPLC-MS system (ACQUITY UPLC I ClASS/SCIEX QTRAP 4500, Waters, Milford, MA, USA). A Phenomenex Kineter EVO C18 (2.1 × 50 mm, 2.6 μm) was used. The temperature was 30 °C. The mobile phase with the flow rate of 0.4 mL·min^−1^ consisted of 0.1% formic acid in water (A) and 0.1% formic acid in methanol (B). The elution was carried out as follows: 20% A at 0–1.0 min; 20% A to 100% A at 1.0–3.0 min; 100% A at 3.0–4.0 min; 100% A to 20%A at 4.0–4.10 min, 20% A at 4.1–7 min. The injection volume was 10 μL and the partial loop with needle overfill mode was used for sample injection. The mass spectrometer was operated in negative ionization mode using MRM to assess the R1: *m*/*z* 931.9→637.5 for R1 and *m*/*z* 779.7→617.4 for saikosaponin (IS). The optimized cone voltage and collision energy were 295 V and 52 eV for R1, 290 V and 48 eV for IS, respectively. A spray voltage of 4500 V was used, and the capillary temperature was 550 °C. The scanning range was selected as *m*/*z* 100→1200: no interference was observed around target compound peaks. Data acquisition and processing were accomplished on a 4500 Q TRAP® mass spectrometer (Applied Biosystems, Foster, CA, USA).

#### 3.6.2. Chromatographic System and Conditions of SAs

The HPLC system consisted of a LC-10AT series quaternary LC pump from Shimadzu Technology (Kyoto, Japan), with a SPD-10 AVP ultraviolet detector, CTO-10ASV column oven, SCL-10AVP controller and CLASS-VP work station. The mobile phase consisted of methanol: acetonitrile (3:1) (A) and 0.3% phosphoric acid in water (B). The flow rate was kept at 1.0 mL·min^−1^. The system was run with a gradient program of 83% B to 60% B at 0–10 min, 60% B to 50% B at 10–20 min, 50% B to 35% B at 20–30 min, 35% B to 83% B at 30–31 min, 83% B at 31–41 min. The detection wavelength was 286 nm.

### 3.7. Preparation of Calibration Standard and Quality Control Samples

#### 3.7.1. Notoginsenoside R1

Calibration standard of R1 at concentrations of 1.97, 5.15, 10.6, 18.2, 51.9, 94.8, 211, 505, and 964 ng·mL^−1^ were prepared by spiking the appropriate amount of saikosaponin A (50 ng·mL^−1^) standard solutions in blank plasma obtained from healthy rats. Similarly, quality control samples (QC) at low concentration (6 ng·mL^−1^), medium concentration (60 ng·mL^−1^) and high concentration (800 ng·mL^−1^) were also prepared as described.

#### 3.7.2. Salvianolic Acids 

Calibration standard of SA-B at concentrations of 0.19, 0.38, 0.76, 1.52, 3.04, 6.08, 12.16 µg·mL^−1^ were prepared in blank plasma obtained from healthy rats. Similarly, quality control samples (QC) at low concentration (0.19 µg·mL^−1^), medium concentration (0.76 µg·mL^−1^) and high concentration (3.04 µg·mL^−1^) were also prepared as described. Calibration standard of SA-A at concentrations of 0.058, 0.12, 0.23, 0.47, 0.93, 1.87, 3.74, 7.48, 14.95, 29.9 µg·mL^−1^ were prepared in blank plasma obtained from healthy rats. Similarly, quality control samples (QC) at low concentration (7.48 µg·mL^−1^), medium concentration (3.74 µg·mL^−1^) and high concentration (0.93 µg·mL^−1^) were also prepared as described. Calibration standard of rosmarinic acid at concentrations of 0.055, 0.11, 0.22, 0.44, 0.89, 1.78, 3.55, 7.1, 14.2, 28.4 µg·mL^−1^ were prepared in blank plasma obtained from healthy rats. Similarly, quality control samples (QC) at low concentration (7.1 µg·mL^−1^), medium concentration (3.55 µg·mL^−1^) and high concentration (0.89 µg·mL^−1^) were also prepared as described.

### 3.8. Preparation of Blood Samples

#### 3.8.1. Notoginsenoside R1

Ten μL saikosaponin A (50 ng·mL^−1^ in acetonitrile, IS) was added to 100 μL of blank blood. Then, the analytical sample was vortexed with 0.39 mL acetonitrile for 5 min and centrifuged at 14,000 r × min^−1^ for 10 min. The supernatant was collected, transferred to a clean centrifuge tube, and evaporated to dryness using a vacuum centrifugal thickener (Centrivap, LABCONCO, Kansas, MO, USA). The resulting residue was dissolved in 100 μL 20% acetonitrile, vortexed for 3 min and centrifuged at 14,000 r·min^−1^ for 10 min for twice, and injected into UPLC-MS/MS system for analysis.

#### 3.8.2. Salvianolic Acids

Thirty μL hydrochloric acid (2 mol·L^−1^) was added to 200 μL of blank blood. Then, the analytical sample was vortexed with 2 mL ethyl acetate for 90 s and centrifuged at 12,000 r·min^−1^ for 5 min. The supernatant was collected, transferred to a clean centrifuge tube, and evaporated to dryness. The resulting residue was dissolved in 100 μL methanol, vortexed for 20 s and centrifuged at 12,000 r·min^−1^ for 5 min, and injected into HPLC system for analysis.

### 3.9. Method Validation

#### 3.9.1. Specificity

The specificity of the method was investigated by analyzing blood from rats, spiked blood samples and blood samples after oral administration of drugs, to exclude any endogenous co-eluting interference.

#### 3.9.2. Recovery Rate

The extraction recovery at three different concentrations (low, medium and high concentrations) was determined. The absolute recovery was determined in three replicates by comparing the peak areas of the extracted samples to the peak areas obtained from the organic solutions at the same concentration.

#### 3.9.3. Precision

The precision of the method was determined by analyzing six replicates at low, medium and high concentrations with the same analytical run on three consecutive days, respectively.

#### 3.9.4. Lower Limit of Quantitation (LOQ)

The LOQ of R1, RA, SA-B and SA-A under chromatographic conditions were determined at an S/N (Signal/Noise) of about 10, respectively

### 3.10. Administration of Drugs Containing SNAC

Male Sprague-Dawley rats with a mean body weight of 250 ± 20 g were randomly divided into 6 groups (*n* = 6). The animals were fasted 12 h prior to oral administration and had free access to water during the experiment. All groups were treated by oral administration via oral gavage and dosing volume of 10 mL·kg^−1^. R1 was oral administrated with 200 mg·kg^−1^. SAs was oral administrated with 500 mg·kg^−1^. Blood samples of 200 μL were withdrawn from the eye socket according to the specific time intervals of at 0, 0.25, 0.5, 1, 1.5, 2, 4, 6, 8, 12 h after administration and immediately mixed with 10 μL of 1% heparin sodium to prevent clotting. Blood samples were centrifuged (10 min, 3000 rpm) and plasma was collected and stored at 20 °C until analysis.

### 3.11. Data Processing

Pharmacokinetics parameters were processed by non-compartmental analysis using the Phoenix WinNonlin 6.0 (Pharsight, Princeton, NJ, USA). Statistical analyses were performed using SPSS16.0. Statistical significance was considered to be reached at *p* < 0.05.

## 4. Conclusions

In this article, a novel absorption enhancer (SNAC) was synthesized and its absorption enhancement effect on drugs with different polarity properties were studied. Caco-2 cell transport in vitro and pharmacokinetics in vivo experiments were conducted. The administration of SAs and R1 in the presence of SNAC resulted in an enhancement of membrane transport. TEER values during the exposure of Caco-2 cells to SNAC suggested that the absorption-enhancing effect of SNAC may not involve tight junction complex opening. The in vivo results demonstrated that SNAC could promote oral absorption of both R1 and SAs. What’s more, it could prolong the retention time in vivo and promote oral absorption of polar molecules to a greater extent. The possible mechanism is ions interact to form complexes between the polar molecules and SNAC. The current work demonstrates the capability of SNAC to increase the epithelial permeability of small hydrophilic molecules, especially for polar molecules.

## Figures and Tables

**Figure 1 molecules-23-02990-f001:**
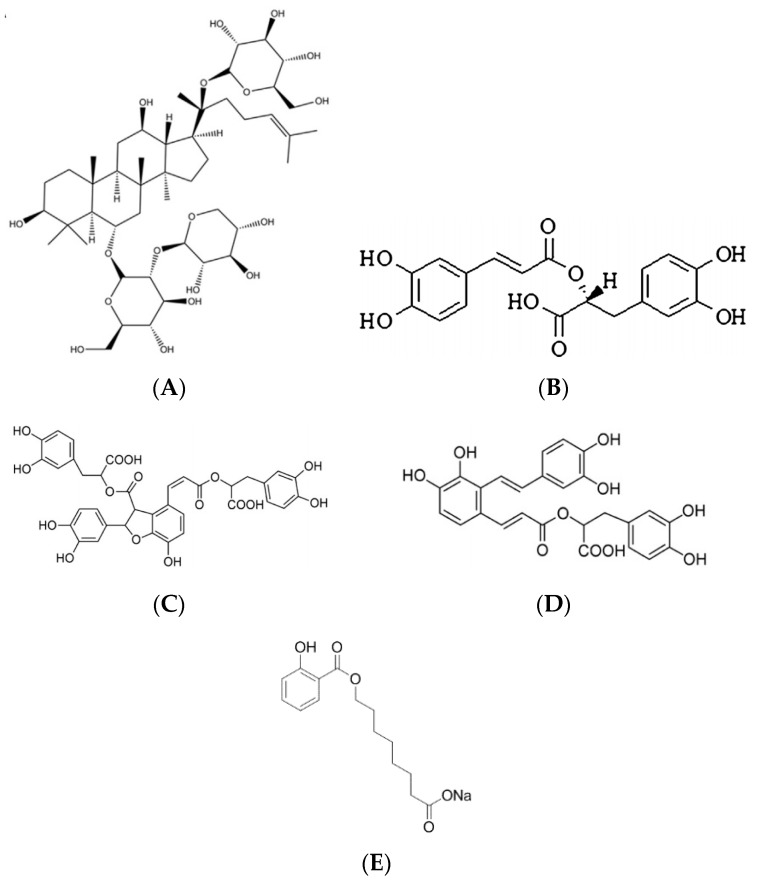
The chemical structures of sodium *N*-[8-(2-hydroxybenzoyl)amino]caprylate (SNAC) and drugs. (**A**) notoginsenoside R1. (**B**) rosmarinic acid. (**C**) salvianolic acid B. (**D**) salvianolic acid A. (**E**) SNAC.

**Figure 2 molecules-23-02990-f002:**
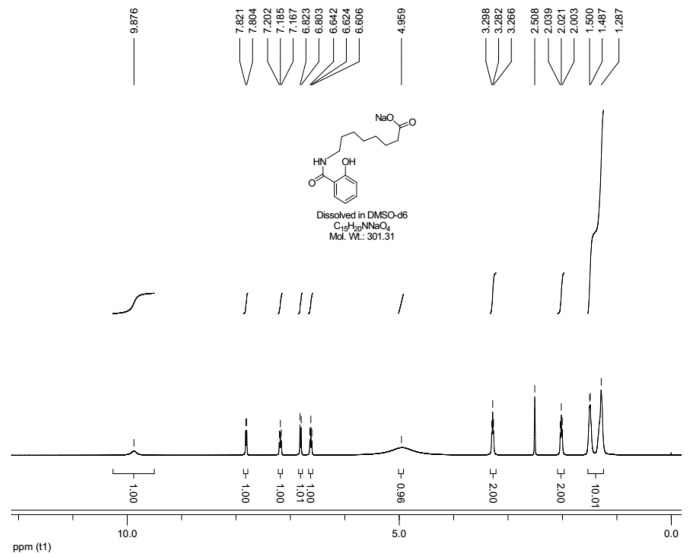
^1^H-NMR characterization of SNAC.

**Figure 3 molecules-23-02990-f003:**
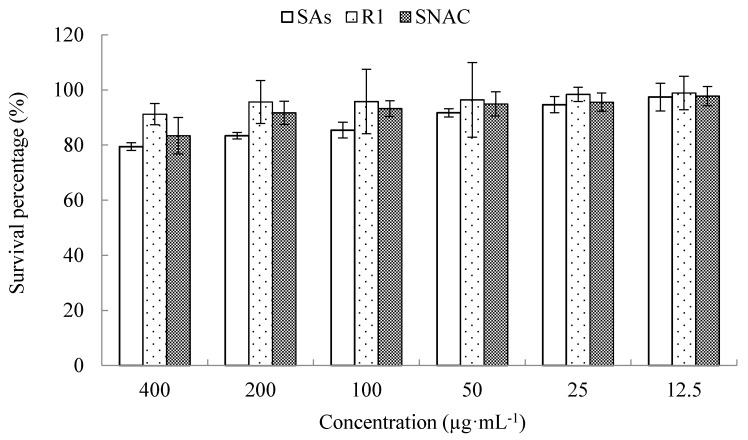
Caco-2 cells were pre-incubated with notoginsenoside R1, Salvianolic acids and Sodium N-[8-(2-hydroxybenzoyl)amino]caprylate (SNAC) for 24 h, and cell viability was assessed by the MTT assay. Values were represented as mean ± SD (*n* = 6, each group).

**Figure 4 molecules-23-02990-f004:**
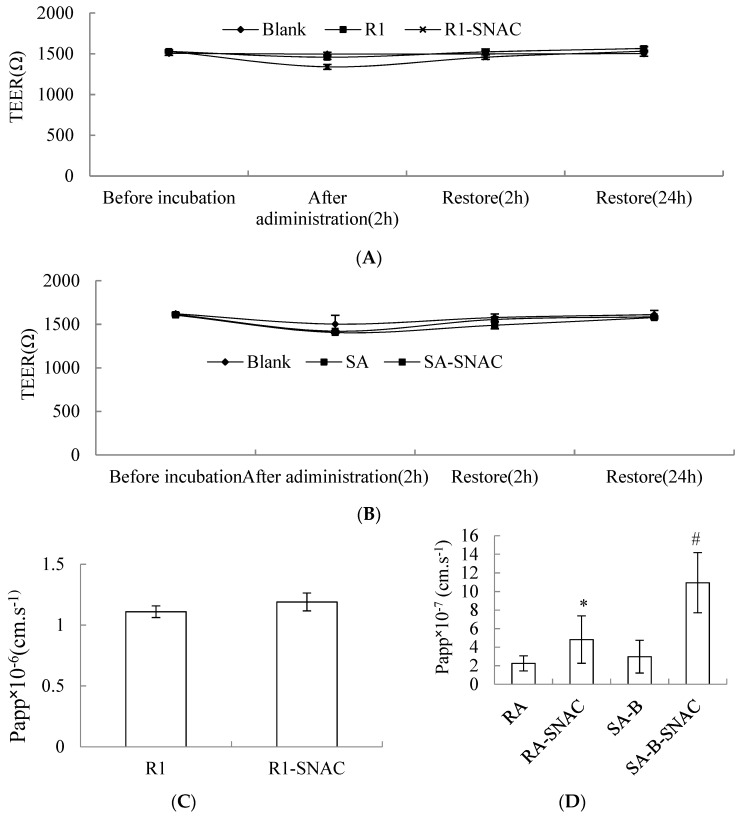
The transport across Caco-2 cell monolayer. (**A**) The transepithelial electrical resistance (TEER) change of Caco-2 cell monolayer treated with notoginsenoside R1 (R1) and R1 containing Sodium N-[8-(2-hydroxybenzoyl)amino]caprylate (SNAC). (**B**) TEER change of Caco-2 cell monolayer treated with salvianolic acids (SAs) and SAs containing SNAC. (**C**) Papp of R1 transport across Caco-2 cell monolayer. (**D**) Papp of rosmarinic acid (RA) and salvianolic acid B (SA-B) of SAs transport across Caco-2 cell monolayer. Values were represented as mean ± SD (*n* = 3, each group). * *p* < 0.05 vs. RA group. # *p* < 0.05 vs. SA-B group. Note: 200 μg·mL^−1^ for R1; 50 μg·mL^−1^ for SAs; R1 or SAs: SNAC (1:1).

**Figure 5 molecules-23-02990-f005:**
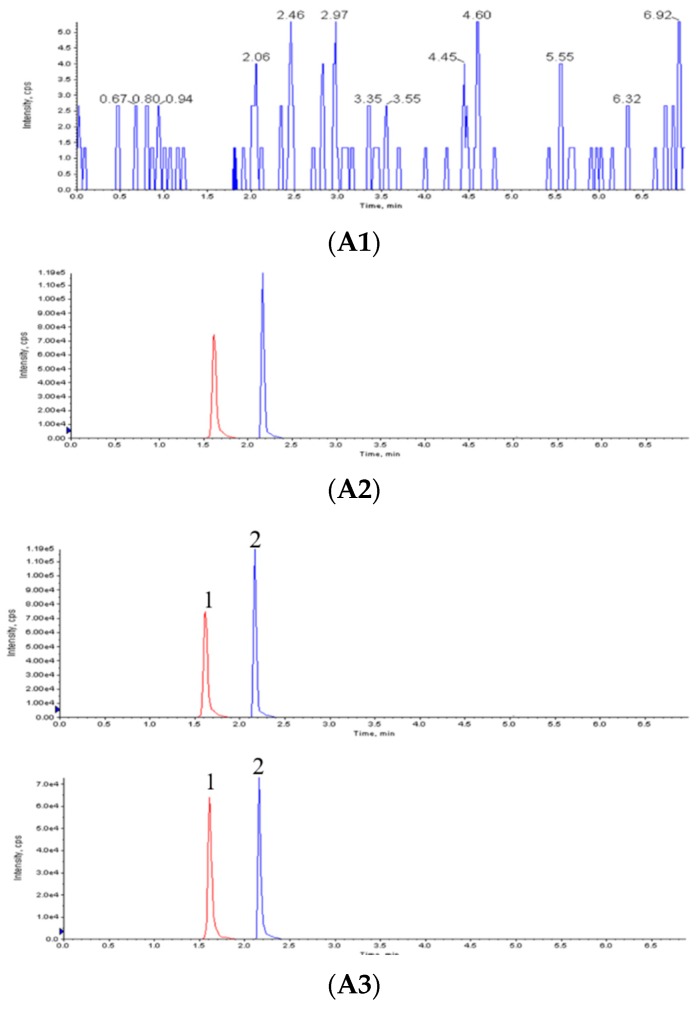
Chromatogram chart. (**A**) UPLC-MS/MS chromatograms of notoginsenoside R1 (R1). Blank rat plasma (**A1**); Rat plasma spiked with R1 (**A2**); Plasma sample after oral administration of R1 containing Sodium N-[8-(2-hydroxybenzoyl)amino]caprylate (SNAC) (**A3**). 1. R1, 2. IS. (**B**) HPLC chromatograms of salvianolic acids (SAs). Blank Blood (**B1**); Rat plasma spiked with mixture the standard solution (**B2**); Plasma sample after oral administration of SAs containing SNAC (**B3**). 1. Rosmarinic acid, 2. Salvianolic acid B, 3. Salvianolic acid A.

**Figure 6 molecules-23-02990-f006:**
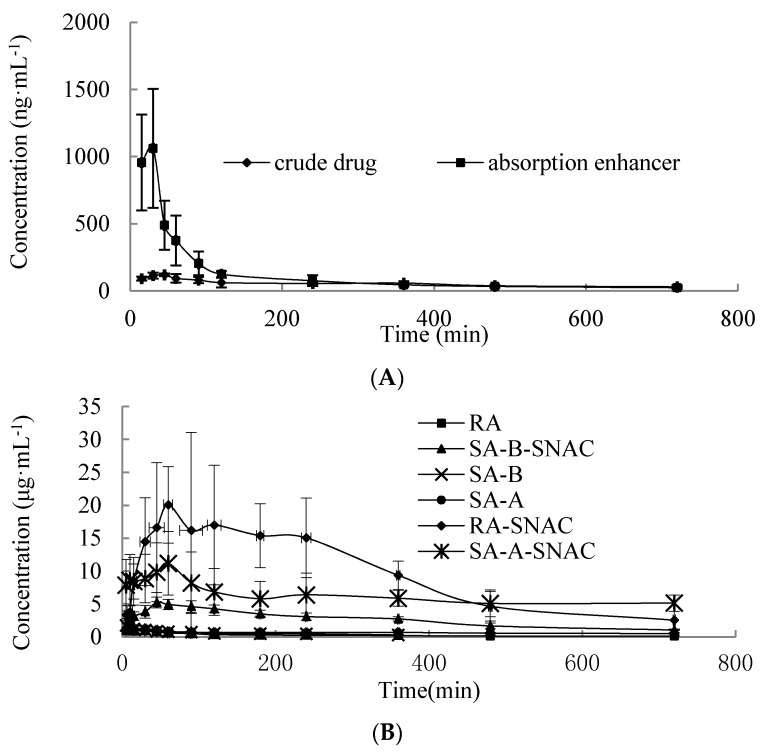
The concentration-time curve of drugs in blood after oral administration of notoginsenoside R1 (**A**) and salvianolic acids (**B**). Values were represented as mean ± SD (*n* = 6, each group).

**Table 1 molecules-23-02990-t001:** Pharmacokinetic parameters of R1 after oral administration of R1 and R1 containing SNAC (*n* = 6).

Parameter	R1	R1 containing SNAC
Cmax (ng·mL^−1^)	119.40 ± 9.86	1061.40 ± 443.60
Tmax (min)	42.50 ± 11.29	20.00 ± 7.74
AUC_0→t_/(ng/mL·min)	37,991.01 ± 2746.71	84,930.61 ± 14,364.23
Fr/%	100	223.55

**Table 2 molecules-23-02990-t002:** Pharmacokinetic parameters of RA, SA-B and SA-A after oral administration of SAs and SAs containing SNAC (*n* = 6).

Parameter	RA	RA Containing SNAC	SA-B	SA-B Containing SNAC	SA-A	SA-A Containing SNAC
Cmax (µg·mL^−1^)	1.02 ± 0.51	20.06 ± 5.79	2.29 ± 0.77	5.62 ± 0.85	1.61 ± 0.44	11.19 ± 4.84
Tmax (min)	10 ± 5	63.75 ± 18.87	7.5 ± 2.74	57.5 ± 17.54	8.75 ± 2.5	67.5 ± 15
AUC_0→t_/(µg/mL·min)	193.79 ± 51.78	6835.35 ± 946.52	214.46 ± 37.86	1871.38 ± 424.47	478.54 ± 28.93	4416.15 ± 706.94
Fr/%	100	3527.21	100	872.30	100	922.83

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
