# Peer review of "Impact of Sodium N-[8-(2-Hydroxybenzoyl)amino]-caprylate on Intestinal Permeability for Notoginsenoside R1 and Salvianolic Acids in Caco-2 Cells Transport and Rat Pharmacokinetics"

_molecules, 2018, doi:10.3390/molecules23112990_

Round 1

Reviewer 1 Report

Major: The paper is poorly organized. There are no statistical methods or descriptions of data presented in several of the figures.

1.       There is no mention of number of wells evaluated for the MTT assay, or filters evaluated for the TEER and Papp evaluations. The figure legends should indicate this too. There is no mention of how the data are presented for these studies. Average (mean or median?) +/- what? The figure legends should also indicate this.

2.       The paper spends an inordinate amount of results, methods and discussion on the purity determination and quantitation of R1 and SAs. The title needs to be changed to reflect this emphasis, or the emphasis needs to change by deemphasizing the characterization of SAs and R1, and focus more on the Caco-2 and rat PK results. Tables 1 and 2 can be combined. Sections 2.4 to 2.7 can be moved to the Methods section.

Minor:

1.       Introduction section

a.       Need references for:

                                                               i.      Page 2 lines 46-49 (2nd sentence of paragraph)

                                                             ii.      Sentence beginning line 50 on page 2, and ending on line 53 needs references for each molecule (insulin, calcitonin, heparin).

                                                           iii.      Page 2, sentence beginning on line 63 and ending on line 66.

                                                           iv.      Page 2, last sentence of paragraph that begins on line 74. If this is the opinion of the authors, that the bioavailability is low due to poor membrane penetration, then they should state this.

b.       Sentence beginning on Line 77:  “Sas and R1 were chosen as model drugs.” Please indicate which class SAs and R1 belong to (polar charged or non-polar charged).

2.       Methods section

a.       Line 292 on page 11. Change adsorption to absorption.

b.       Section 3.7.2: the concentrations of standards for SA-A and the ZC samples need to be corrected. Same applies for rosmarinic acid.

c.       3.8.2: 30 uL of 2 mol/L HCl?  Do you mean in HCl? What was the concentration of the HCl?

3.       Results section

a.       Figure 3 legend: indicate that incubation time was 24 hours.

b.       Figure 4 should be split into two figures: one figure for TEER and one for Papp. Figure 4 D as currently shown is not labeled with “D”. TEER is not a measure of permeability. As used in this work, it is a measure of monolayer integrity.

c.       Figure 4A: What was the concentration of SAs, R1, SNAC? These should be indicated in the figure legend and in the text in Section 2.3. Methods section 3.5 indicates these concentrations. However, Methods states that SAs concentration was 50 ug/mL, but only one result is shown. Which SA (-A, -B, rosmarinic acid)?

d.       Figure 4D: Why are there no results shown for SA-A?

e.       Section 2.8 for PK analysis. What test was used for Tmax change analysis (line 214)?

f.        Paragraph beginning on Line 219 in Section 2.8:  What does “It” refer to?

Reviewer 2 Report

The quality of figures, tables should be improved.

Figure 1. The structural formulas too small, the specific functional groups in some places not visible clearly. It should be increased the size of formulas.

Figure 2. The figure of HNMR spectrum should be increased. The HPLC and LCMS-UPLC chromatograms scale and peak legend should be increased.

Figure 5. The figure caption is not clear. Above the figure in "2.4.1.UPLC-MS/MS Chromatogram" subchapter the authors refer to Figure 2 A, Figure 2B-a and b, Figure 2B-c and d. As I see it should be Figure 5 A and Figure  5B-a, -b-c and -d. Below the Figure 5 the caption is wrong, uppercase and lowercase sign is not correct. This part of the manuscript is very confused.

Figure 6. the legends  are slipped onthe peek.

Table 3. and 4. Parameter information and units should be writen in same row, then the information would be clear.

Other informations: materials and methods clear, reproducible and the result explanation, discussion are correct.

Round 2

Reviewer 1 Report

No further changes are needed to the revised manuscript.